# Provable Submodular Minimization using Wolfe's Algorithm

**Deeparnab Chakrabarty**[*]  **Prateek Jain**[*]  **Pravesh Kothari**[†]

## Abstract

Owing to several applications in large scale learning and vision problems, fast submodular function minimization (SFM) has become a critical problem. Theoretically, unconstrained SFM can be performed in polynomial time [10, 11]. However, these algorithms are typically not practical. In 1976, Wolfe [21] proposed an algorithm to find the minimum Euclidean norm point in a polytope, and in 1980, Fujishige [3] showed how Wolfe's algorithm can be used for SFM. For general submodular functions, this Fujishige-Wolfe minimum norm algorithm seems to have the best empirical performance.

Despite its good practical performance, very little is known about Wolfe's minimum norm algorithm theoretically. To our knowledge, the only result is an exponential time analysis due to Wolfe [21] himself. In this paper we give a maiden convergence analysis of Wolfe's algorithm. We prove that in $t$ iterations, Wolfe's algorithm returns an $O(1/t)$-approximate solution to the min-norm point on *any* polytope. We also prove a robust version of Fujishige's theorem which shows that an $O(1/n^2)$-approximate solution to the min-norm point on the base polytope implies *exact* submodular minimization. As a corollary, we get the first pseudo-polynomial time guarantee for the Fujishige-Wolfe minimum norm algorithm for unconstrained submodular function minimization.

## 1 Introduction

An integer-valued[1] function $f : 2^X \to \mathbb{Z}$ defined over subsets of some finite ground set $X$ of $n$ elements is submodular if it satisfies the following *diminishing marginal returns* property: for every $S \subseteq T \subseteq X$ and $i \in X \setminus T$, $f(S \cup \{i\}) - f(S) \geq f(T \cup \{i\}) - f(T)$. Submodularity arises naturally in several applications such as image segmentation [17], sensor placement [18], etc. where minimizing an arbitrary submodular function is an important primitive.

In submodular function minimization (SFM), we assume access to an *evaluation oracle for $f$* which for any subset $S \subseteq X$ returns the value $f(S)$. We denote the time taken by the oracle to answer a single query as EO. The objective is to find a set $T \subseteq X$ satisfying $f(T) \leq f(S)$ for every $S \subseteq X$. In 1981, Grotschel, Lovasz and Schrijver [8] demonstrated the first polynomial time algorithm for SFM using the ellipsoid algorithm. This algorithm, however, is practically infeasible due to the running time and the numerical issues in implementing the ellipsoid algorithm. In 2001, Schrijver [19] and Iwata et al. [9] independently designed *combinatorial* polynomial time algorithms for SFM. Currently, the best algorithm is by Iwata and Orlin [11] with a running time of $O(n^5 \text{EO} + n^6)$.

However, from a practical stand point, none of the provably polynomial time algorithms exhibit good performance on instances of SFM encountered in practice (see §4). This, along with the widespread applicability of SFM in machine learning, has inspired a large body of work on *practically* fast procedures (see [1] for a survey). But most of these procedures focus either on special submodular

---

[*]Microsoft Research, 9 Lavelle Road, Bangalore 560001.

[†]University of Texas at Austin (Part of the work done while interning at Microsoft Research)

[1]One can assume any function is integer valued after suitable scaling.

functions such as decomposable functions [16, 20] or on constrained SFM problems [13, 12, 15, 14].

**Fujishige-Wolfe's Algorithm for SFM:** For any submodular function $f$, the *base polytope* $\mathcal{B}_f$ of $f$ is defined as follows:

$$\mathcal{B}_f = \{x \in \mathbb{R}^n : x(A) \le f(A), \ \forall A \subset X, \ \text{and} \ x(X) = f(X)\}, \tag{1}$$

where $x(A) := \sum_{i \in A} x_i$ and $x_i$ is the $i$-th coordinate of $x \in \mathbb{R}^n$. Fujishige [3] showed that if one can obtain the minimum norm point on the base polytope, then one can solve SFM. Finding the minimum norm point, however, is a non-trivial problem; at present, to our knowledge, the only polynomial time algorithm known is via the ellipsoid method. Wolfe [21] described an iterative procedure to find minimum norm points in polytopes as long as linear functions could be (efficiently) minimized over them. Although the base polytope has exponentially many constraints, a simple greedy algorithm can minimize any linear function over it. Therefore using Wolfe's procedure on the base polytope coupled with Fujishige's theorem becomes a natural approach to SFM. This was suggested as early as 1984 in Fujishige [4] and is now called the Fujishige-Wolfe algorithm for SFM.

This approach towards SFM was revitalized in 2006 when Fujishige and Isotani [6, 7] announced encouraging computational results regarding the minimum norm point algorithm. In particular, this algorithm significantly out-performed all known *provably* polynomial time algorithms. Theoretically, however, little is known regarding the convergence of Wolfe's procedure except for the finite, but exponential, running time Wolfe himself proved. Nor is the situation any better for its application on the base polytope. Given the practical success, we believe this is an important, and intriguing, theoretical challenge.

In this work, we make some progress towards analyzing the Fujishige-Wolfe method for SFM and, in fact, Wolfe's algorithm in general. In particular, we prove the following two results:

- We prove (in Theorem 4) that for *any* polytope $\mathcal{B}$, Wolfe's algorithm converges to an $\varepsilon$-approximate solution, in $O(1/\varepsilon)$ steps. More precisely, in $O(nQ^2/\varepsilon)$ iterations, Wolfe's algorithm returns a point $\|x\|_2^2 \le \|x_*\|_2^2 + \varepsilon$, where $Q = \max_{p \in \mathcal{B}} \|p\|_2$.

- We prove (in Theorem 5) a robust version of a theorem by Fujishige [3] relating min-norm points on the base polytope to SFM. In particular, we prove that an approximate min-norm point solution provides an approximate solution to SFM as well. More precisely, if $x$ satisfies $\|x\|_2^2 \le z^T x + \varepsilon^2$ for all $z \in \mathcal{B}_f$, then, $f(S_x) \le \min_S f(S) + 2n\varepsilon$, where $S_x$ can be constructed efficiently using $x$.

Together, these two results gives us our main result which is a pseudopolynomial bound on the running time of the Fujishige-Wolfe algorithm for submodular function minimization.

**Theorem 1. (Main Result.)** *Fix a submodular function $f : 2^X \to \mathbb{Z}$. The Fujishige-Wolfe algorithm returns the minimizer of $f$ in $O((n^5\text{EO} + n^7)F^2)$ time where $F := \max_{i=1}^n (|f(\{i\})|, |f([n]) - f([n] \setminus i)|)$.*

Our analysis suggests that the Fujishige-Wolfe's algorithm is dependent on $F$ and has worse dependence on $n$ than the Iwata-Orlin [11] algorithm. To verify this, we conducted empirical study on several standard SFM problems. However, for the considered benchmark functions, running time of Fujishige-Wolfe's algorithm seemed to be independent of $F$ and exhibited better dependence on $n$ than the Iwata-Orlin algorithm. This is described in §4.

## 2 Preliminaries: Submodular Functions and Wolfe's Algorithm

### 2.1 Submodular Functions and SFM

Given a ground set $X$ on $n$ elements, without loss of generality we think of it as the first $n$ integers $[n] := \{1, 2, \ldots, n\}$. $f$ be a submodular function. Since submodularity is translation invariant, we assume $f(\emptyset) = 0$. For a submodular function $f$, we write $\mathcal{B}_f \subseteq \mathbb{R}^n$ for the associated base polyhedron of $f$ defined in (1). Given $x \in \mathbb{R}^n$, one can find the minimum value of $q^\top x$ over $q \in \mathcal{B}_f$ in $O(n \log n + n\text{EO})$ time using the following greedy algorithm: Renumber indices such that $x_1 \le \cdots \le x_n$. Set $q_i^* = f([i]) - f([i-1])$. Then, it can be proved that $q^* \in \mathcal{B}_f$ and is the minimizer of the $x^\top q$ for $q \in \mathcal{B}_f$.

The connection between the SFM problem and the base polytope was first established in the following minimax theorem of Edmonds [2].

**Theorem 2** (Edmonds [2]). *Given any submodular function $f$ with $f(\emptyset) = 0$, we have*

$$\min_{S \subseteq [n]} f(S) = \max_{x \in \mathcal{B}_f} \left( \sum_{i:x_i < 0} x_i \right)$$

The following theorem of Fujishige [3] shows the connection between finding the minimum norm point in the base polytope $\mathcal{B}_f$ of a submodular function $f$ and the problem of SFM on input $f$. This forms the basis of Wolfe's algorithm. In §3.2, we prove a robust version of this theorem.

**Theorem 3** (Fujishige's Theorem [3]). *Let $f : 2^{[n]} \to \mathbb{Z}$ be a submodular function and let $\mathcal{B}_f$ be the associated base polyhedron. Let $x^*$ be the optimal solution to $\min_{x \in \mathcal{B}_f} ||x||$. Define $S = \{i \mid x_i^* < 0\}$. Then, $f(S) \leq f(T)$ for every $T \subseteq [n]$.*

## 2.2 Wolfe's Algorithm for Minimum Norm Point of a polytope.

We now present Wolfe's algorithm for computing the minimum-norm point in an arbitrary polytope $\mathcal{B} \subseteq \mathbb{R}^n$. We assume a *linear optimization oracle* (LO) which takes input a vector $x \in \mathbb{R}^n$ and outputs a vector $q \in \arg\min_{p \in \mathcal{B}} x^\top p$.

We start by recalling some definitions. The *affine hull* of a finite set $S \subseteq \mathbb{R}^n$ is $\mathtt{aff}(S) = \{y \mid y = \sum_{z \in S} \alpha_z \cdot z, \sum_{z \in S} \alpha_z = 1\}$. The *affine minimizer* of $S$ is defined as $y = \arg\min_{z \in \mathtt{aff}(S)} ||z||_2$, and $y$ satisfies the following *affine minimizer property*: for any $v \in \mathtt{aff}(S)$, $v^\top y = ||y||^2$. The procedure $\mathtt{AffineMinimizer}(S)$ returns $(y, \alpha)$ where $y$ is the affine minimizer and $\alpha = (\alpha_s)_{s \in S}$ is the set of coefficients expressing $y$ as an affine combination of points in $S$. This procedure can be naively implemented in $O(|S|^3 + n|S|^2)$ as follows. Let $B$ be the $n \times |S|$ matrix where each column in a point in $S$. Then $\alpha = (B^\top B)^{-1} \mathbf{1}/\mathbf{1}^\top (B^\top B)^{-1} \mathbf{1}$ and $y = B\alpha$.

---

**Algorithm 1** Wolfe's Algorithm

1. Let $q$ be an arbitrary vertex of $\mathcal{B}$. Initialize $x \leftarrow q$. We always maintain $x = \sum_{i \in S} \lambda_i q_i$ as a convex combination of a subset $S$ of vertices of $\mathcal{B}$. Initialize $S = \{q\}$ and $\lambda_1 = 1$.
2. **WHILE** (`true`): (MAJOR CYCLE)
   (a) $q := \mathrm{LO}(x)$.                                           *// Linear Optimization: $q \in \arg\min_{p \in \mathcal{B}} x^\top p$.*
   (b) **IF** $||x||^2 \leq x^\top q + \varepsilon^2$ **THEN** `break`.           *// Termination Condition. Output $x$.*
   (c) $S := S \cup \{q\}$.
   (d) **WHILE** (`true`) : (MINOR CYCLE)
       i. $(y, \alpha) = \mathtt{AffineMinimizer}(S)$.                  *//$y = \arg\min_{z \in \mathtt{aff}(S)} ||z||$.*
       ii. **IF** $\alpha_i \geq 0$ for all $i$ **THEN** `break`.         *//If $y \in \mathtt{conv}(S)$, then end minor loop.*
       iii. **ELSE**
           *// If $y \notin \mathtt{conv}(S)$, then update $x$ to the intersection of the boundary of $\mathtt{conv}(S)$ and the segment joining $y$ and*
           *previous $x$. Delete points from $S$ which are not required to describe the new $x$ as a convex combination.*
           $\theta := \min_{i:\alpha_i < 0} \lambda_i/(\lambda_i - \alpha_i)$                    *// Recall, $x = \sum_i \lambda_i q_i$.*
           Update $x \leftarrow \theta y + (1-\theta)x$.              *// By definition of $\theta$, the new $x$ lies in $\mathtt{conv}(S)$.*
           Update $\lambda_i \leftarrow \theta\alpha_i + (1-\theta)\lambda_i$.       *//This sets the coefficients of the new $x$*
           $S = \{i : \lambda_i > 0\}$.                 *// Delete points which have $\lambda_i = 0$. This deletes at least one point.*
   (e) Update $x \leftarrow y$.     *// After the minor loop terminates, $x$ is updated to be the affine minimizer of the current set $S$.*
3. **RETURN** $x$.

---

When $\varepsilon = 0$, the algorithm on termination (if it terminates) returns the minimum norm point in $\mathcal{B}$ since $||x||^2 \leq x^\top x_* \leq ||x|| \cdot ||x_*||$. For completeness, we sketch Wolfe's argument in [21] of finite termination. Note that $|S| \leq n$ always; otherwise the affine minimizer is 0 which either terminates the program or starts a minor cycle which decrements $|S|$. Thus, the number of minor cycles in a major cycle $\leq n$, and it suffices to bound the number of major cycles. Each major cycle is associated with a set $S$ whose affine minimizer, which is the current $x$, lies in the convex hull of $S$. Wolfe calls such sets *corrals*. Next, we show that $||x||$ strictly decreases across iterations (major or minor cycle) of the algorithm, which proves that no corral repeats, thus bounding the number of major cycles by the number of corrals. The latter is at most $\binom{N}{n}$, where $N$ is the number of vertices of $\mathcal{B}$.

Consider iteration $j$ which starts with $x_j$ and ends with $x_{j+1}$. Let $S_j$ be the set $S$ at the beginning of iteration $j$. If the iteration is a major cycle, then $x_{j+1}$ is the affine minimizer of $S_j \cup \{q_j\}$

where $q_j = \mathrm{LO}(x_j)$. Since $x_j^\top q_j < ||x_j||^2$ (the algorithm doesn't terminate in iteration $j$) and $x_{j+1}^\top q_j = ||x_{j+1}||^2$ (affine minimizer property), we get $x_j \neq x_{j+1}$, and so $||x_{j+1}|| < ||x_j||$ (since the affine minimizer is unique). If the iteration is a minor cycle, then $x_{j+1} = \theta x_j + (1-\theta)y_j$, where $y_j$ is the affine minimizer of $S_j$ and $\theta < 1$. Since $||y_j|| < ||x_j||$ ($y_j \neq x_j$ since $y_j \notin \mathtt{conv}(S_j)$), we get $||x_{j+1}|| < ||x_j||$.

## 3 Analysis

Our refined analysis of Wolfe's algorithm is encapsulated in the following theorem.

**Theorem 4.** *Let $\mathcal{B}$ be an* arbitrary *polytope such that the* maximum *Euclidean norm of any vertex of $\mathcal{B}$ is at most $Q$. After $O(nQ^2/\varepsilon^2)$ iterations, Wolfe's algorithm returns a point $x \in \mathcal{B}$ which satisfies $||x||^2 \leq x^\top q + \varepsilon^2$, for all points $q \in \mathcal{B}$. In particular, this implies $||x||^2 \leq ||x_*||^2 + 2\varepsilon^2$.*

The above theorem shows that Wolfe's algorithm converges to the minimum norm point at an $1/t$-rate. We stress that the above is for *any* polytope. To apply this to SFM, we prove the following robust version of Fujishige's theorem connecting the minimum norm point in the base polytope and the set minimizing the submodular function value.

**Theorem 5.** *Fix a submodular function $f$ with base polytope $\mathcal{B}_f$. Let $x \in \mathcal{B}_f$ be such that $||x||^2 \leq x^\top q + \varepsilon^2$ for all $q \in \mathcal{B}_f$. Renumber indices such that $x_1 \leq \cdots \leq x_n$. Let $S = \{1, 2, \ldots, k\}$,where $k$ is smallest index satisfying (C1) $x_{k+1} \geq 0$ and (C2) $x_{k+1} - x_k \geq \varepsilon/n$. Then, $f(S) \leq f(T) + 2n\varepsilon$ for any subset $T \subseteq S$. In particular, if $\varepsilon = \frac{1}{4n}$ and $f$ is integer-valued, then $S$ is a minimizer.*

Theorem 4 and Theorem 5 implies our main theorem.

**Theorem 1. (Main Result.)** *Fix a submodular function $f : 2^X \to \mathbb{Z}$. The Fujishige-Wolfe algorithm returns the minimizer of $f$ in $O((n^5\mathrm{EO} + n^7)F^2)$ time where $F := \max_{i=1}^n \left(|f(\{i\})|, |f([n]) - f([n] \setminus i)|\right)$.*

*Proof.* The vertices of $\mathcal{B}_f$ are well understood: for every permutation $\sigma$ of $[n]$, we have a vertex with $x_{\sigma(i)} = f(\{\sigma(1), \ldots, \sigma(i)\}) - f(\{\sigma(1), \ldots, \sigma(i-1)\})$. By submodularity of $f$, we get for all $i$, $|x_i| \leq F$. Therefore, for any point $x \in \mathcal{B}_f$, $||x||^2 \leq nF^2$. Choose $\varepsilon = 1/4n$. From Theorem 4 we know that if we run $O(n^4F^2)$ iterations of Wolfe, we will get a point $x \in \mathcal{B}_f$ such that $||x||^2 \leq x^\top q + \varepsilon^2$ for all $q \in \mathcal{B}_f$. Theorem 5 implies this solves the SFM problem. The running time for each iteration is dominated by the time for the subroutine to compute the affine minimizer of $S$ which is at most $O(n^3)$, and the linear optimization oracle. For $\mathcal{B}_f$, $\mathrm{LO}(x)$ can be implemented in $O(n\log n + n\mathrm{EO})$ time. This proves the theorem. $\qquad\square$

We prove Theorem 4 and Theorem 5 in §3.1 and §3.2, respectively.

### 3.1 Analysis of Wolfe's Min-norm Point Algorithm

The stumbling block in the analysis of Wolfe's algorithm is the interspersing of major and minor cycles which oscillates the size of $S$ preventing it from being a good measure of progress. Instead, in our analysis, we use the norm of $x$ as the measure of progress. Already we have seen that $||x||$ strictly decreases. It would be nice to quantify how much the decrease is, say, across one major cycle. This, at present, is out of our reach even for major cycles which contain two or more minor cycles in them. However, we can prove significant drop in norm in major cycles which have at most one minor cycle in them. We call such major cycles *good*. The next easy, but very useful, observation is the following: one cannot have too many bad major cycles without having too many good major cycles.

**Lemma 1.** *In any consecutive $3n+1$ iterations, there exists at least one good major cycle.*

*Proof.* Consider a run of $r$ iterations where all major cycles are bad, and therefore contain $\geq 2$ minor cycles. Say there are $k$ major cycles and $r - k$ minor cycles, and so $r - k \geq 2k$ implying $r \geq 3k$. Let $S_I$ be the set $S$ at the start of these iterations and $S_F$ be the set at the end. We have $|S_F| \leq |S_I| + k - (r - k) \leq |S_I| + 2k - r \leq n - \frac{r}{3}$. Therefore, $r \leq 3n$, since $|S_F| \geq 0$. $\qquad\square$

Before proceeding, we introduce some notation.

**Definition 1.** *Given a point $x \in \mathcal{B}$, let us denote $\mathtt{err}(x) := ||x||^2 - ||x_*||^2$. Given a point $x$ and $q$, let $\Delta(x, q) := ||x||^2 - x^\top q$ and let $\Delta(x) := \max_{q \in \mathcal{B}} \Delta(x, q) = ||x||^2 - \min_{q \in \mathcal{B}} x^\top q$. Observe that $\Delta(x) \geq \mathtt{err}(x)/2$ since $\Delta(x) \geq ||x||^2 - x^\top x_* \geq (||x||^2 - ||x_*||^2)/2$.*

We now use $t$ to index all good major cycles. Let $x_t$ be the point $x$ at the beginning of the $t$-th good major cycle. The next theorem shows that the norm significantly drops across good major cycles.

**Theorem 6.** *For $t$ iterating over good major cycles, $\mathtt{err}(x_t) - \mathtt{err}(x_{t+1}) \geq \Delta^2(x_t)/8Q^2$.*

We now complete the proof of Theorem 4 using Theorem 6.

**Proof of Theorem 4.** Using Theorem 6, we get that $\mathtt{err}(x_t) - \mathtt{err}(x_{t+1}) \geq \mathtt{err}(x_t)^2/32Q^2$ since $\Delta(x) \geq \mathtt{err}(x)/2$ for all $x$. We claim that in $t^* \leq 64Q^2/\varepsilon^2$ good major cycles, we reach $x_t$ with $\mathtt{err}(x_{t^*}) \leq \varepsilon^2$. To see this rewrite as follows:

$$\mathtt{err}(x_{t+1}) \leq \mathtt{err}(x_t)\left(1 - \frac{\mathtt{err}(x_t)}{32Q^2}\right), \quad \text{for all } t.$$

Now let $e_0 := \mathtt{err}(x_0)$. Define $t_0, t_1, \ldots$ such that for all $k \geq 1$ we have $\mathtt{err}(x_t) > e_0/2^k$ for $t \in [t_{k-1}, t_k)$. That is, $t_k$ is the first time $t$ at which $\mathtt{err}(x_t) \leq e_0/2^k$. Note that for $t \in [t_{k-1}, t_k)$, we have $\mathtt{err}(x_{t+1}) \leq \mathtt{err}(x_t)\left(1 - \frac{e_0}{32Q^2 2^k}\right)$. This implies in $32Q^2 2^k/e_0$ time units after $t_{k-1}$, we will have $\mathtt{err}(x_t) \leq \mathtt{err}(x_{t_{k-1}})/2$; we have used the fact that $(1-\delta)^{1/\delta} < 1/2$ when $\delta < 1/32$. That is, $t_k \leq t_{k-1} + 32Q^2 2^k/e_0$. We are interested in $t^* = t_K$ where $2^K = e_0/\varepsilon^2$. We get $t^* \leq \frac{32Q^2}{e_0}\left(1 + 2 + \cdots + 2^K\right) \leq 64Q^2 2^K/e_0 = 64Q^2/\varepsilon^2$.

Next, we claim that in $t^{**} < t^* + t'$ good major cycles, where $t' = 8Q^2/\varepsilon^2$, we obtain an $x_{t^{**}}$ with $\Delta(x_{t^{**}}) \leq \varepsilon^2$. This is because, if not, then, using Theorem 6, in each of the good major cycles $t^* + 1, t^* + 2, \ldots t^* + t'$, $\mathtt{err}(x)$ falls additively by $> \varepsilon^4/8Q^2$ and thus $\mathtt{err}(x_{t^*+t'}) < \mathtt{err}(x_{t^*}) - \varepsilon^2 \leq 0$, which is a contradiction. Therefore, in $O(Q^2/\varepsilon^2)$ good major cycles, the algorithm obtains an $x = x_{t^{**}}$ with $\Delta(x) \leq \varepsilon^2$, proving Theorem 4. □

The rest of this subsection is dedicated to proving Theorem 6.

**Proof of Theorem 6:** We start off with a simple geometric lemma.

**Lemma 2.** *Let $S$ be a subset of $\mathbb{R}^n$ and suppose $y$ is the minimum norm point of $\mathtt{aff}(S)$. Let $x$ and $q$ be* arbitrary *points in $\mathtt{aff}(S)$. Then,*

$$||x||^2 - ||y||^2 \geq \frac{\Delta(x,q)^2}{4Q^2} \tag{2}$$

*where $Q$ is an upper bound on $||x||, ||q||$.*

*Proof.* Since $y$ is the minimum norm point in $\mathtt{aff}(S)$, we have $x^\top y = q^\top y = ||y||^2$. In particular, $||x - y||^2 = ||x||^2 - ||y||^2$. Therefore,

$$\Delta(x,q) = ||x||^2 - x^T q = ||x||^2 - x^\top y + y^\top q - x^T q = (y - x)^T(q - x) \leq ||y - x|| \cdot ||q - x||$$
$$\leq ||y - x||(||x|| + ||q||) \leq 2Q||y - x||,$$

where the first inequality is Cauchy-Schwartz and the second is triangle inequality. Lemma now follows by taking square of the above expression and by observing that $||y - x||^2 = ||x||^2 - ||y||^2$. □

The above lemma takes case of major cycles with no minor cycles in them.

**Lemma 3** (Progress in Major Cycle with no Minor Cycles)**.** *Let $t$ be the index of a good major cycle with no minor cycles. Then $\mathtt{err}(x_t) - \mathtt{err}(x_{t+1}) \geq \Delta^2(x_t)/4Q^2$.*

*Proof.* Let $S_t$ be the set $S$ at start of the $t$th good major cycle, and let $q_t$ be the point minimizing $x_t^\top q$. Let $S = S_t \cup q_t$ and let $y$ be the minimum norm point in $\mathtt{aff}(S)$. Since there are no minor cycles, $y \in \mathtt{conv}(S)$. Abuse notation and let $x_{t+1} = y$ be the iterate at the call of the next major cycle (and not the next good major cycle). Since the norm monotonically decreases, it suffices to prove the lemma statement for this $x_{t+1}$. Now apply Lemma 2 with $x = x_t$ and $q = q_t$ and $S = S_t \cup q_t$. We have that $\mathtt{err}(x_t) - \mathtt{err}(x_{t+1}) = ||x_t||^2 - ||y||^2 \geq \Delta(x_t, q_t)^2/4Q^2 = \Delta(x_t)^2/4Q^2$. □

Now we have to argue about major cycles with exactly one minor cycle. The next observation is a useful structural result.

**Lemma 4** (New Vertex Survives a Minor Cycle.). *Consider any (not necessarily good) major cycle. Let $x_t, S_t, q_t$ be the parameters at the beginning of this cycle, and let $x_{t+1}, S_{t+1}, q_{t+1}$ be the parameters at the beginning of the next major cycle. Then, $q_t \in S_{t+1}$.*

*Proof.* Clearly $S_{t+1} \subseteq S_t \cup q_t$ since $q_t$ is added and then maybe minor cycles remove some points from $S$. Suppose $q_t \notin S_{t+1}$. Well, then $S_{t+1} \subseteq S_t$. But $x_{t+1}$ is the affine minimizer of $S_{t+1}$ and $x_t$ is the affine minimizer of $S_t$. Since $S_t$ is the larger set, we get $||x_t|| \leq ||x_{t+1}||$. This contradicts the strict decrease in the norm. $\square$

**Lemma 5** (Progress in an iteration with exactly one minor cyvle). *Suppose the tth good major cycle has exactly one minor cycle. Then, $\texttt{err}(x_t) - \texttt{err}(x_{t+1}) \geq \Delta(x_t)^2/8Q^2$.*

*Proof.* Let $x_t, S_t, q_t$ be the parameters at the beginning of the $t$th good major cycle. Let $y$ be the affine minimizer of $S_t \cup q_t$. Since there is one minor cycle, $y \notin \texttt{conv}(S_t \cup q_t)$. Let $z = \theta x_t + (1-\theta)y$ be the intermediate $x$, that is, point in the line segment $[x_t, y]$ which lies in $\texttt{conv}(S_t \cup q_t)$. Let $S'$ be the set after the single minor cycle is run. Since there is just one minor cycle, we get $x_{t+1}$ (abusing notation once again since the next major cycle maynot be good) is the affine minimizer of $S'$.

Let $A \triangleq ||x_t||^2 - ||y||^2$. From Lemma 2, and using $q_t$ is the minimizer of $x_t^\top q$ over all $q$, we have:

$$A = ||x_t||^2 - ||y||^2 \geq \Delta^2(x_t)/4Q^2 \tag{3}$$

Recall, $z = \theta x_t + (1-\theta)y$ for some $\theta \in [0,1]$. Since $y$ is the min-norm point of $\texttt{aff}(S_t \cup q_t)$, and $x_t \in S_t$, we get $||z||^2 = \theta^2||x_t||^2 + (1-\theta^2)||y||^2$. this yields:

$$||x_t||^2 - ||z||^2 = (1-\theta^2)\left(||x_t||^2 - ||y||^2\right) = (1-\theta^2)A \tag{4}$$

Further, recall that $S'$ is the set after the only minor cycle in the $t^{th}$ iteration is run and thus, from Lemma 4, $q_t \in S'$. $z \in \texttt{conv}(S')$ by definition. And since there is only one minor cycle, $x_{t+1}$ is the affine minimizer of $S'$. We can apply Lemma 2 with $z, q_t$ and $x_{t+1}$, to get

$$||z||^2 - ||x_{t+1}||^2 \geq \frac{\Delta^2(z, q_t)}{4Q^2} \tag{5}$$

Now we lower bound $\Delta^2(z, q_t)$. By definition of $z$, we have:

$$z^\top q_t = \theta x_t^\top q_t + (1-\theta)y^\top q_t = \theta x_t^\top q_t + (1-\theta)||y||^2$$

where the last equality follows since $y^\top q_t = ||y||^2$ (since $q_t \in S_t \cup q_t$ and $y$ is affine minimizer of $S_t \cup q_t$). This gives

$$
\begin{aligned}
\Delta(z, q_t) &= ||z||^2 - z^\top q_t \\
&= \left(\theta^2||x_t||^2 + (1-\theta^2)||y||^2\right) - \left(\theta x_t^\top q_t + (1-\theta)||y||^2\right) \\
&= \theta(||x_t||^2 - x_t^\top q_t) - \theta(1-\theta)\left(||x_t||^2 - ||y||^2\right) \\
&= \theta\left(\Delta(x_t) - (1-\theta)A\right)
\end{aligned}
\tag{6}
$$

From (4),(5), and (6), we get

$$\texttt{err}_t - \texttt{err}_{t+1} \geq (1-\theta^2)A + \frac{\theta^2\left(\Delta(x_t) - (1-\theta)A\right)^2}{4Q^2} \tag{7}$$

We need to show that the RHS is at least $\Delta(x_t)^2/8Q^2$. Intuitively, if $\theta$ is small (close to 0), the first term implies this using (3), and if $\theta$ is large (close to 1), then the second term implies this. The following paragraph formalizes this intuition for any $\theta$.

Now, if $(1-\theta^2)A > \Delta(x_t)^2/8Q^2$, we are done. Therefore, we assume $(1-\theta^2)A \leq \Delta(x_t)^2/8Q^2$. In this case, using the fact that $\Delta(x_t) \leq ||x_t||^2 + ||x_t||||q_t|| \leq 2Q^2$, we get that

$$(1-\theta)A \leq (1-\theta^2)A \leq \Delta(x_t) \cdot \frac{\Delta(x_t)}{8Q^2} \leq \Delta(x_t)/4$$

Substituting in (7), and using (3), we get

$$\texttt{err}_t - \texttt{err}_{t+1} \geq \frac{(1-\theta^2)\Delta(x_t)^2}{4Q^2} + \frac{9\theta^2\Delta(x_t)^2}{64Q^2} \geq \frac{\Delta(x_t)^2}{8Q^2} \tag{8}$$

This completes the proof of the lemma. $\square$

Lemma 3 and Lemma 5 complete the proof of Theorem 6.

### 3.2   A Robust version of Fujishige's Theorem

In this section we prove Theorem 5 which we restate below.

**Theorem 5.** *Fix a submodular function $f$ with base polytope $\mathcal{B}_f$. Let $x \in \mathcal{B}_f$ be such that $||x||^2 \leq x^\top q + \varepsilon^2$ for all $q \in \mathcal{B}_f$. Renumber indices such that $x_1 \leq \cdots \leq x_n$. Let $S = \{1, 2, \ldots, k\}$, where $k$ is smallest index satisfying (C1) $x_{k+1} \geq 0$ and (C2) $x_{k+1} - x_k \geq \varepsilon/n$. Then, $f(S) \leq f(T) + 2n\varepsilon$ for any subset $T \subseteq S$. In particular, if $\varepsilon = \frac{1}{4n}$ and $f$ is integer-valued, then $S$ is a minimizer.*

Before proving the theorem, note that setting $\varepsilon = 0$ gives Fujishige's theorem Theorem 3.

*Proof.* We claim that the following inequality holds. Below, $[i] := \{1, \ldots, i\}$.

$$\sum_{i=1}^{n-1} (x_{i+1} - x_i) \cdot (f([i]) - x([i])) \leq \varepsilon^2 \tag{9}$$

We prove this shortly. Let $S$ and $k$ be as defined in the theorem statement. Note that $\sum_{i \in S: x_i \geq 0} x_i \leq n\varepsilon$, since (C2) doesn't hold for any index $i < k$ with $x_i \geq 0$. Furthermore, since $x_{k+1} - x_k \geq \varepsilon/n$, we get using (9), $f(S) - x(S) \leq n\varepsilon$. Therefore, $f(S) \leq \sum_{i \in S: x_i < 0} x_i + 2n\varepsilon$ which implies the theorem due to Theorem 2.

Now we prove (9). Let $z \in \mathcal{B}_f$ be the point which minimizes $z^\top x$. By the Greedy algorithm described in Section 2.1, we know that $z_i = f([i]) - f([i-1])$. Next, we write $x$ in a different basis as follows: $x = \sum_{i=1}^{n-1} (x_i - x_{i+1}) \mathbf{1}_{[i]} + x_n \mathbf{1}_{[n]}$. Here $\mathbf{1}_{[i]}$ is used as the shorthand for the vector which has 1's in the first $i$ coordinates and 0s everywhere else. Taking dot product with $(x - z)$, we get

$$||x||^2 - x^\top z = (x - z)^\top x = \sum_{i=1}^{n-1} (x_i - x_{i+1}) \left( x^\top \mathbf{1}_{[i]} - z^\top \mathbf{1}_{[i]} \right) + x_n \left( x^\top \mathbf{1}_{[n]} - z^\top \mathbf{1}_{[n]} \right) \tag{10}$$

Since $z_i = f([i]) - f([i-1])$, we get $x^\top \mathbf{1}_{[i]} - z^\top \mathbf{1}_{[i]}$ is $x([i]) - f([i])$. Therefore the RHS of (10) is the LHS of (9). The LHS of (10), by the assumption of the theorem, is at most $\varepsilon^2$ implying (9). $\square$

## 4   Discussion and Conclusions

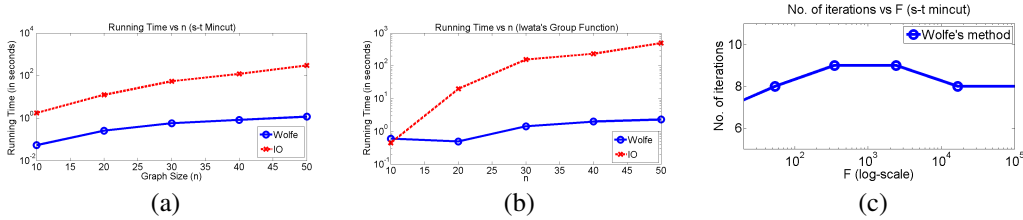

(a)                              (b)                              (c)

Figure 1: Running time comparision of Iwata-Orlin's (IO) method [11] vs Wolfe's method. (a): s-t mincut function, (b) Iwata's 3 groups function [16]. (c): Total number of iterations required by Wolfe's method for solving s-t mincut with increasing $F$

We have shown that the Fujishige-Wolfe algorithm solves SFM in $O((n^5 \mathrm{EO} + n^7)F^2)$ time, where $F$ is the maximum change in the value of the function on addition or deletion of an element. Although this is the first pseudopolynomial time analysis of the algorithm, we believe there is room for improvement and hope our work triggers more interest.

Note that our anlaysis of the Fujishige-Wolfe algorithm is weaker than the best known method in terms of time complexity (IO method by [11]) on two counts: a) dependence on $n$, b) dependence on $F$. In contrast, we found this algorithm significantly outperforming the IO algorithm empirically – we show two plots here. In Figure 1 (a), we run both on Erdos-Renyi graphs with $p = 0.8$ and randomly chosen $s, t$ nodes. In Figure 1 (b), we run both on the Iwata group functions [16] with 3 groups. Perhaps more interestingly, in Figure 1 (c), we ran the Fujishige-Wolfe algorithm on the simple path graph where $s, t$ were the end points, and changed the capacities on the edges of the graph which changed the parameter $F$. As can be seen, the number of iterations of the algorithm remains constant even for exponentially increasing $F$.

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
