[Reviews · NeurIPS 2014]

Submitted by Assigned_Reviewer_13

In this paper, the authors give several theoretical analyses on the convergence of the minimum-norm-point (MNP) algorithm, which is known as the practically fastest algorithm for submodular minimization. An analysis on the convergence of the MNP algorithm is one of the most important open problems on submodular optimization. Since I don't fully understand the details of the paper, I give a few minor comments, assuming that the theoretical results are correct (that seems to be ok after reading the other reviewers).

- The convergence result would be the first one for the MNP algorithm, which would be great. Meanwhile the obtained complexity is worse than the known best one for submodular minimization. Although, as the authors claim, the MNP algorithm is known to be faster in practice, it doesn’t seem to mean very much to me how it is useful.

- The authors show the empirical comparison with the IO method in the method. However, it doesn't seem to have much meaning for the discussion of this paper because it is well known that the MNP works faster than the IO method in practice (there are many papers to have much more diverse empirical examples). It seems to be better not to include it or to carry out ones more related to the discussions in the paper.
Summary: The authors give several theoretical analyses on the convergence of the minimum-norm-point (MNP) algorithm. I have just a few minor comments to the authors.

Submitted by Assigned_Reviewer_26

This paper analyses Wolfe's algorithm to prove that it finds a O(1/t)-approximate solution to the min-norm point.
The paper has two major contributions:

1) The first pseudo-polynomial time guarantee for the Fujishige-Wolfe minimum norm algorithm for submodular function minimization.

2) The first pseudo-polynomial time guarantee for the Fujishige-Wolfe algorithm

I only fleetingly verified the proofs, they seem OK.
Summary: New analysis of Wolfe's algorithm for submodular minimization with new results, a nice contribution to this line of research.

Submitted by Assigned_Reviewer_27

This paper investigates the minimum norm point algorithm for submodular minimization, and shows the first pseudo polynomial time computational guarantee for the minimum norm point algorithm.

This observation is novel, and the proof technique seems correct to me. I feel that understanding the running time of the minimum norm point algorithm is an important problem for both the machine learning and theory communities, and this paper provides a first investigation of this.

I feel this paper should be accepted, though the worst case running time is rather pessimistic. On the negative side, I would have a loved to see more empirical investigation on the kind of functions where the minimum norm point algorithm is slow. The authors only provide simulations for the s-t mincut function. It would also be interesting to see some investigation and possible directions on what the lower bound on the complexity might look like.
Summary: This paper provides the first investigation on the running time of the minimum norm point algorithm. The proof techniques seem right to me, though I haven't carefully checked all of them.

I think this paper makes a lot of contribution to the submodular optimization community, and after discussions, we feel this paper is a strong accept.

Submitted by Assigned_Reviewer_36

Your analysis of the Wolfe's algorithm doesn't exploit any special properties of submodular functions; it applies to any polytope. This was not very clear to me until I've read the proof. Please emphasize this fact in the abstract and in the introduction.

When stating complexities of algorithms (yours and others) please include the number of required oracle calls (EO); this is a common practice for submodular minimization algorithms.

087: B => B_f

134: "we will assume that we have a linear optimization oracle". This is a very strange assumption which actually never holds; only sorting requires O(n \log n), and then you need to call the oracle n times.

140: should it be O(r^3 + nr^2) (for computing S^T S)? It would be useful for the reader to explain how you view S^T as a matrix and state its dimension and the dimensions of vectors 1 in the formula.

Starting with section 2.2, the polytope is sometimes denoted as B and sometimes as P (e.g. in 146 and 251). Please be consistent.

Algorithm 1:
- I believe the stopping criterion in line 3 is wrong - it should be the same as in Theorem 3(b)
- line 3(a): S\cup q => S\cup\{q\}
- line 3(c).iii: Z:={i: ...} => Z:={q_i: ...}
- Suggestion: lines 2 and 3(d) have identical operations. You may want to restructure it as follows:
WHILE true (MAJOR CYCLE)
(a) q:= ...
(b) if (stopping condition) break (or terminate)
(c) S := S\cup \{q\}
...

Then it will be more clear that every major cycle starts by calling the optimization oracle.
Siimlarly for the minor loop: (y,\alpha)=AffineMinimizer(S) can be the first step of the minor loop.
Also, in beginning of the minor loop you could some comments, such
- compute maximum $\theta$ such that $x_\theta=\theta y + (1-\theta) x \in conv(S)$
- express $x_\theta$ as a convex combination of points from S, remove points with zero weight

This would have saved me quite some time.

Set $S$ in the algorithm has a very different meaning from the set $S$ in eq. (1) (and other similar places). Could you use different letters?

278: I think the comma should go before "for any $v\in S$", not after. Also, I don't think you need a comma in "good, major cycle" (though I am not a native English speaker).

322: Capitalize "this".

347: Please add $(1-\theta) A \le (1-\theta^2) A \le ..." for reader's convenience.

Experiments: please discuss (and ideally compare) the family of problems used in [16]. In the conclusions [16] says that "the minimum norm algorithm with high accuracy is much slower than the other methods".
Summary: The authors prove the first pseudo-polynomial bound for the complexity of Wolfe's algorithm for computing a minimum norm over general polytopes. (According to the paper, the only previous bound was by Wolfe himself, and it was exponential). The authors then apply this to the Fujishige's algorithm for submodular minimization (which minimizes the norm over the base polyhedron of a submodular function). With an extra (not very surpising) lemma they thus prove a pseudo-polynomial bound for the Fujishige's algorithm.

It is impressive that the authors managed to make a progress on a problem that dates back to 1976. On the negative side, the proposed analysis doesn't really shed any light on the good performance of the Fujishige's algorithm (in my view); the submodularity property is not exploited in any way in the proposed analysis of the Wolfe's algorithm.

Submitted by Assigned_Reviewer_42

The present paper gives a version of the Fujishige-Wolfe
algorithm for submodular function minimization and shows its
pseudo-polynomial time complexity.
The Fujishige-Wolfe algorithm has been widely used and runs
efficiently in practice, but the problem of determining its
algorithmic complexity has been open until now.
The pseudo-polynomiality result of the present paper is the
first big step toward determining its exact complexity.

Comments (mostly minor ones) are listed below.
Major ones are Comments 11 and 13. Comments 1 and 8 are medium.
(Three-digit numbers denote the corresponding lines in the
text.)

----------------------------------------------

1. 055-056: Iwata and Orlin's algorithm [11] is not the
fastest. Orlin's algorithm published in Mathematical
Programming, Ser.A, Vol.118 (2009), pp.237-251 is currently the
fastest and runs in O(n^5EO + n^6) time where EO denotes the
time for the function evaluation oracle.

2. 066: Here should be mentioned the relation between $n$ and
$X$, which will appear on Line 105.

3. 093: $F$ should be defined here.

4. 095: `the Wolfe's' should be read `the Wolfe' or `Wolfe's.'

5. 107: `for any two' should be read `for all two' (within the
``if'' clause).

6. 109: `to find the set' should be read `to find a set.'

7. 138: `for a subset' should be read `for an affinely
independent subset.'

8. 176: The inequality at the end of this line can be
$N_{\rm min} \le N_{\rm maj}$ without factor $n$,
since $N_{\rm maj}-N_{\rm min} \ge {\rm the dimension of the final $S$}.

9. 178: The factor $n$ can be removed (due to the previous
comment).

10. 213-214: The comment starting with ``Observe that …'' is not
precise (or invalid), which should be removed. It is actually
irrelevant to the present proof.

11. 263: I cannot see $t=32Q^2/\epsilon^2$, but can only see
$t=32Q^4/\epsilon^4$, which is much larger than the former.
(Please check it carefully. I could be wrong. Any short proof
of it would be useful for readers.) This would affect the
complexity estimation all through the rest.
Also the present $t$ and the one appearing before have
different meanings; the two should be distinguished.

12. 316: `(abusing notation) is' should be read `(abusing
notation) that is.'

13. 359 (Theorem 7): When $2n\epsilon < 1$, we can find
a set $S$ in Theorem 7 without any oracle call by setting
$S=\{ i \mid x_i < -1/(2n) \}$.
(Compare the rounding technique proposed in Fujishige and
Isotani's paper [7]. (See the remark below Theorem 3.3 and
equations (3.11) and (3.12) on Page 7 in [7]. Note that we also
have $||x - x_*|| \le \epsilon$.)
Theorem 7 itself is still correct.

14. 375: `at least $\epsilon^2$' should be read `at least
$-\epsilon^2$.'

15. 409: Check the complexity. See Comment 11.

----------------------------------------------------------------
Summary: I found this paper very interesting and worth publishing.
Author Feedback
Author rebuttal: We thank all the reviewers for detailed and helpful reviews.

Reviewer_13
Significance of results: Provable algorithms for submodular minimization are still very slow in practice, even for simple problems like graph mincut. Hence, we believe that there is a need to develop new and faster algorithms for submodular minimization. As min. norm point (MNP) based algorithms are fast in practice, we believe a better understanding of these methods may lead us to faster poly-time algorithms for submodular minimization. To the best of our knowledge, our work is the first pseudo-poly time analysis of the celebrated Wolfe's algorithm for the MNP problem.

Experiments: Actually, the point of the experiments, as we mention in the paper, is to indicate weakness in our analysis and motivate further study of Wolfe's algorithm since it shows promise in practice and the observed time complexity appears to not depend on the actual values of the function.

Reviewer_27
The paper was meant to give a theoretical reason why the MNP algorithm works well in practice. We agree that the current bound may be too weak, but as far as we know, no theoretical analysis was known prior to this work. We agree with the reviewer that detailed experimental evaluation of MNP algorithm is a good idea. Apart from the s,t- mincut, we did some experiments on the Iwata and group-Iwata function. MNP algorithms only seemed to work better on them than the performance predicted by our analysis.

Reviewer_36
Thanks for suggesting notation and other changes in presentation. We will incorporate them in the future versions.

134 "linear optimization oracle": we meant that we assume an oracle that can optimize any linear function over the polytope "B" (we call this oracle "linear optimization oracle"). We didn't mean that the oracle itself has a linear time complexity.

140 S: S is a nxr matrix. Thanks for pointing out the time complexity bug.

Algorithm 1 issues: Thanks for pointing out bug in the stopping condition and other fixes. Affineminimizer(S) has to be executed before the minor loop; if all the coefficients (alpha) are positive than we need not go into minor loop.

Experiments: We will include more details regarding the function of [16]. [16] compares MNP to certain other more specialized algorithms that do not apply to general submodular minimization.

Reviewer_42
Thanks for the comments, we will fix the typos and take the comments in account while preparing the next draft.

8 and 9: We can have n minor cycles between any two major cycles. Hence, N_{min}\leq n N_{maj}

11 Q^2/eps^2 bound): Using (2), we have err(t)-err(t+1)\geq err(t)^2/32Q^2. Now err(t) is monotonically decreasing. Also, let err(T)=\epsilon, then err(t)\geq \epsilon for t\leq T. Hence, err(t)-err(t+1)\geq \epsilon^2/32Q^2 for all t < T. The bound now follows easily.

13 (Theorem 7, oracle calls): Thanks for pointing this out.
Although, to apply technique of [7], we might need a robust version of their Theorem 3.3, which states that if we are close enough to the min-norm point then we are done. Arguably, this could be easier than Thm 7 in our paper; however, our theorem is probably more "first-principled" and keeps the paper self-contained.